# Kekulé Counts, Clar Numbers, and ZZ Polynomials for All Isomers of (5,6)-Fullerenes C_52_–C_70_

**DOI:** 10.3390/molecules29174013

**Published:** 2024-08-24

**Authors:** Henryk A. Witek, Rafał Podeszwa

**Affiliations:** 1Department of Applied Chemistry, National Yang Ming Chiao Tung University, Hsinchu 300093, Taiwan; 2Institute of Molecular Science, National Yang Ming Chiao Tung University, Hsinchu 300093, Taiwan; 3Institute of Chemistry, University of Silesia in Katowice, Szkolna 9, 40-006 Katowice, Poland

**Keywords:** fullerene isomers, isomer stability, Kekulé count, Clar number, Zhang–Zhang polynomial (aka ZZ polynomial or Clar covering polynomial), bond order

## Abstract

We report an extensive tabulation of several important topological invariants for all the isomers of carbon (5,6)-fullerenes $C_{n}$ with *n* = 52–70. The topological invariants (including Kekulé count, Clar count, and Clar number) are computed and reported in the form of the corresponding Zhang–Zhang (ZZ) polynomials. The ZZ polynomials appear to be distinct for each isomer cage, providing a unique label that allows for differentiation between various isomers. Several chemical applications of the computed invariants are reported. The results suggest rather weak correlation between the Kekulé count, Clar count, Clar number invariants, and isomer stability, calling into doubt the predictive power of these topological invariants in discriminating the most stable isomer of a given fullerene. The only exception is the Clar count/Kekulé count ratio, which seems to be the most important diagnostic discovered from our analysis. Stronger correlations are detected between Pauling bond orders computed from Kekulé structures (or Clar covers) and the corresponding equilibrium bond lengths determined from the optimized DFTB geometries of all 30,579 isomers of C_20_–C_70_.

## 1. Introduction

A (5,6)-fullerene is a polyhedral carbon cage with only pentagonal and hexagonal faces. Each such cage necessarily contains an even number *n* of carbon atoms. The number of pentagonal faces is independent of *n*, always being equal to 12, while the number of hexagonal faces is equal to n/2−10. The relative distributions of the twelve pentagonal faces in the network of the remaining hexagonal faces give rise to a large number of structural isomers of (5,6)-fullerenes (except for n=22, for which no such isomer exists). For the (5,6)-fullerenes studied here, the number of conceivable distinct isomers ranges between 437 for C_52_ to 8149 for C_70_ [1]. Considerable combinatorial effort has been invested in finding algorithms that allow for the generation and enumeration of these isomers. The first solution to this problem was offered by Manolopoulos, Fowler, and their collaborators in the 1990s in the form of a ring spiral algorithm [2,3], in which the structure of each isomer cage is encoded as a linear sequence of pentagons and hexagons. It was soon realized [4,5] that for larger fullerenes the spiral algorithm might miss some of the isomers simply because some of the isomers cannot be encoded as unbranched spiral sequences of pentagons and hexagons. To solve this limitation, Brinkmann and Dress developed a top-down approach [6,7] capable of generating all the isomers of (5,6)-fullerenes $C_{n}$ for a general value of *n*.

The carbon soot obtained in graphite laser vaporization experiments usually contains a mixture of various isomers of carbon clusters with different sizes. Characterization of the soot components and understanding the reasons behind certain fullerene isomers being more abundant in soot compared to others has attracted considerable interest on the part of the chemistry community [8,9,10,11,12,13,14,15,16]. Current beliefs on fullerene isomer stability [17] can be summarized by a number of procedural filters: (i) discard isomers with abutting pentagons; (ii) discard isomers with disparate hexagon neighbour patterns; or (iii) discard isomers with poor electronic structure. Many of these observations can be rationalized using simple geometric arguments [18,19,20] and quantified [21] with semiempirical models based on penalty and merit functions for reappearing motifs (according to which two fused pentagons cost 26.5 kcal mol^−1^ on average, the phenalene motif (i.e., three fused hexagon rings) costs 5.5 kcal mol^−1^ on average, and a pentagon between two hexagons provides a stabilization of 4.5 kcal mol^−1^ on average). A more recent study [22] estimates the pentagon-signature penalty to be on the order of 20–25 kcal mol^−1^. The particularly large penalty for abutting pentagons, usually referred to in the fullerene community as violation of the isolated pentagon rule (IPR) [8], is the most important stability discriminant in the search process for the lowest-energy fullerene isomers. For a perspective on non-IPR isomers, see [23].

A large portion of the predictions and explanations available in the literature is based on topological and graph-theoretical concepts such as aromaticity [20,24,25,26,27,28,29], π-electron resonance energy [30,31,32,33,34], Kekulé structures [35,36,37,38,39,40,41,42], Clar structures [43,44,45,46,47,48,49,50,51,52,53,54,55], and many other related ideas [56,57,58,59,60,61,62,63,64,65,66,67,68,69,70]. The early hypothesis of Kroto et al. [8] stating that the most abundant $I_{h}$ isomer of C_60_ would have a very large number of Kekulé structures (i.e., the Kekulé count K) was soon disproved by Austin and collaborators [35], who found that there are twenty isomers of C_60_ with K larger than that of the $I_{h}$ isomer. Interestingly, a recent DFT study [22] of the thermodynamic stability of all 1812 (5,6)-isomers of C_60_ clearly demonstrated the rather counterintuitive fact that the isomer with the largest value of K is actually the least thermodynamically stable isomer of C_60_ (the value of K= 16,501 for this tubular isomer 60:1 [35,36] is about 30% larger than K= 12,500 for the most-stable $I_{h}$ isomer 60:1812 [30]). Note that the notation n:m corresponds to the m^th^ isomer in the lexicographic spiral order of the fullerene C_*n*_ [3]. In 2010, Zhang and collaborators discovered [71] that the $I_{h}$ isomer of C_60_ indeed maximizes the Kekulé count K, but only among those isomers with the maximal Clar number Cl=8, suggesting a pronounced role of aromaticity in assessing the stability of fullerene isomers (proper mathematical definitions of Cl and K are provided later in this paper; here, we only briefly signalize that the Clar number Cl corresponds to the maximal number of Clar aromatic sextets that can be simultaneously accommodated within a given benzenoid moiety [72]). Zhang and his collaborators found that there are eighteen isomers of C_60_ with Cl=8 [43,48] and that for the $I_{h}$ isomer of C_60_ with K= 12,500, the number of Kekulé structures is distinctly larger than for the next isomer in this class (60:44, with K= 11,259). Another interesting observation of Zhang and his collaborators [71] concerned the tubular isomer 60:1, which indeed maximizes the Kekulé count with K= 16,501, but at the same time (together with five other isomers of C_60_) minimizes the Clar number with Cl=4. The pronounced role of local aromaticity in designating the most stable isomer of C_60_
could perhaps constitute an interesting and valuable tool for characterizing the most stable isomers of a given carbon fullerene without the need for extensive quantum chemical calculations, provided that such a relationship also holds for fullerenes other than C_60_. Unfortunately, the topological descriptors needed for such an analysis have never been reported in the literature.

In a recent article [73], we reported a compilation of topological invariants for all the isomers of small (5,6)-fullerenes C_20_–C_50_, including their Kekulé counts K and Clar numbers Cl. We discovered that for these small and highly-strained fullerenes, the correlation between their thermodynamic stability and their topological invariants is rather disappointing. According to the observations of Zhang et al. for C_60_ [71], the isomer with the maximal Kekulé count K among the isomers with the maximal Clar number Cl should also be the most thermodynamically stable isomer of a given fullerene C_*n*_. However, the data in Figure 4 of [73] show otherwise; the most stable isomer of C_36_ (isomer 36:14 in [3]) with Cl=2 and K=288 corresponds to the minimal Clar number among all the isomers of C_36_, and has an intermediate value of K (the value of K for isomers of C_36_ ranges between 266 and 364). Similarly, while for C_50_, the most stable isomer (50:271 in [3]) with Cl=5 and K=2343 indeed maximizes the Clar number among the isomers of C_50_, its value of K is again intermediate (the value of K for isomers of C_50_ ranges between 2005 and 3276). At this point [73], it remains unclear whether the discrepancy with the previous observations of Zhang et al. [71] made for C_60_ can be attributed to the highly-strained nature of these small fullerene cages, the somewhat exceptional position that C_60_ occupies among all the fullerenes, or perhaps to the coincidental nature of the observations of Zhang et al. [71] Therefore, in the current work we have decided to extend the compilation of topological invariants to all isomers of fullerenes C_52_–C_70_, which should allow us to elucidate the character of the observations made by Zhang et al. [71] for C_60_, and possibly to extend it to larger fullerenes as well. The vast number of isomers treated in this study (29,767 distinct isomers of C_52_–C_70_) prevents us from showing the results of our investigations directly in the main body of the paper. Therefore, the compilation of topological invariants for all isomers of fullerenes C_52_–C_70_ is exiled to the Appendix A accompanying this article (file ZZpolynomials.txt, available at https://www.mdpi.com/article/10.3390/molecules29174013/s1), while the main body of our paper only presents the thermodynamic analysis of isomer stability, the correlation with their topological invariants, and the analysis of bond orders computed on the basis of the developed framework for topological invariants.

The structure of our paper is as follows. Section 2 presents a brief introduction to the computational methods employed in our analysis, including an introduction to the theory of ZZ polynomials used to determine the topological invariants of fullerenes and a brief sketch of the density functional tight binding (DFTB) method used to optimize the isomer structures and compute their energies. Section 3 briefly summarizes the results listed in the Appendix A and explains how they should be interpreted. Section 3.1 and Section 3.2 provide a comparison of the computed thermodynamic stability of fullerene isomers with their various topological invariants, and establish the existence/lack of existence of correlations between these two groups of descriptors. In Section 3.3, we analyze the statistical correlation between two groups of topological invariants (the Clar count C and the Kekulé count K) and discover an interesting regularity related to the most stable isomers and the C/K ratio. Section 3.4 deals with a verification of the hypothesis by Zhang, Ye, and Liu [71] claiming that the most energetically stable structural isomer maximizes the number of Kekulé structures K among the isomers with the maximal conceivable value of Cl. In Section 3.5 we report that all the ZZ polynomials of the 30,579 fullerene isomers studied here are distinct and hence can serve as a convenient unique labels for distinguishing between those isomers. Section 3.6 provides an analysis of the bond orders derived from the topological invariants and their correlation with the actual bond lengths in the optimized fullerene isomers. Finally, Section 4 presents the conclusions of our work.

## 2. Computational Details

The thermodynamic stability of the C_52_–C_70_ fullerene isomers is assessed by computing their total DFTB energies at the equilibrium geometry of each isomer; note that this method has been successfully used for this purpose before [74,75]. Note also that while isomer stability ranking has been performed for a large group of fullerenes (see for example [76,77,78]), the published results usually report only the most stable isomers. Therefore, for the requirements of the current work, we have decided to recompute all the rankings again from scratch while consistently using the same DFTB quantum chemical method. The abbreviation DFTB corresponds to the density-functional tight-binding method [79], an approximate quantum chemical approach unifying the elements of tight-binding methodology with density-functional parameterization of the matrix elements. In DFTB, only the valence electrons of each atom are treated explicitly using a minimal valence basis set. The one-electron Hamiltonian and overlap integrals are precomputed and stored in so-called Slater–Koster (SK) files, whereas the contributions from core electrons and various double-counting terms are included via effective distance-dependent two-center repulsive potentials. The electron repulsion is accounted for via attenuated interaction of self-consistently determined atomic Mulliken charges. Further details of the DFTB methodology can be found in various reviews of the method [80,81,82]. DFTB is often used to model carbon nanostructures, including fullerenes, for which it often shows accuracy comparable with density functional theory (DFT) [83]. As a sanity check, in Figure 1 we show a comparison of the optimized DFT and DFTB energies for all 1812 isomers of C_60_; the DFTB energies and optimized structures are taken from the current work, while the DFT energies and optimized structures are taken from [22]. This comparison shows very good linear correlation between both methods (with $R^{2}=0.993$), with correct identification of the most stable and least stable isomers along with rather small deviations for all of the intermediate isomers. The multiple points located below the correlation line in Figure 1 are invariably associated with lower-energy minima discovered by the DFTB method; these new energy minima often originate from Jahn–Teller distortion of the original cage, which often removes degeneracies in the frontier orbital spectrum and lowers the global symmetry of the isomer (further details are discussed later in this section). In the current work, we use so-called full third-order DFTB together with the 3OB SK parameter file for carbon without any dispersion correction. All calculations are performed using the DFTB+ program [84] with the $1\times 10^{-12}$ convergence criterion for the self-consistent charges, closely following the spirit of our earlier work [16,73,85,86,87]. The atomic force convergence criterion was initially set to $1\times 10^{-7}$; however, this value turned out to be too close to the numerical accuracy limit of the first geometrical DFTB derivatives. Consequently, this convergence criterion has been somewhat loosened and all of the structures have been optimized with the maximal force not larger than $2.3\times 10^{-7}$, an optimization criterion applicable to all the studied here isomers of C_52_–C_70_. To unify the discussion and to allow for comparisons with smaller fullerenes, the same methodology was also extended to the (5,6)-isomers of C_20_–C_50_ treated initially in our previous work [73] and discussed here again for completeness.

In 49 cases (out of 29,767), the fullerene isomers display a distinct open-shell character which prevents convergence of the DFTB calculations to a non-metallic solution. In these cases (denoted as “HOMO-LUMO 0 gap” in the accompanying Appendix A), we have used an electronic temperature of T=0.00001 K and obtained a DFTB solution corresponding to fractional orbital occupations. Almost all of these cases correspond to two quasi-degenerate molecular orbitals and two electrons occupying them; the corresponding HOMO:LUMO fractional occupations range between 1.000:1.000 and 1.906:0.094. In one case (isomer 8148 of C_70_), the frontier HOMO orbital is doubly degenerate and the LUMO orbital is non-degenerate; the corresponding occupation pattern involves four electrons, and can be summarized as 1:504:1.504:0.992. For these open-shell cases, it is more informative to express the stability of a given fullerene isomer via the Mermin free energy; however, in all studied cases the difference with respect to the total energy is smaller than $3\times 10^{-2}$ kcal/mol, meaning that it has no practical significance. In any event, the DFTB solutions with fractional occupations are not rigorous and do not correspond to a well-defined quantum spin number, and as such should be treated with caution. The electronic spin state in the previous DFT study of all the isomers of C_60_ was selected on the base of simple Hückel calculations [22]. A large proportion of isomers of C_60_ (231 out of 1812, about 12.7%) corresponded to triplet (or even quintet) electronic states in these calculations, spin states that would be overlooked by the open-shell DFTB calculations reported here. Surprisingly, in our DFTB calculations only one isomer of C_60_ out of 1812 (60:1478) has a manifestly open-shell structure with a zero HOMO-LUMO gap, while only four other isomers (60:1535, 60:1574, 60:1374, and 60:1481) have HOMO:LUMO gaps smaller than 1 eV. Evidently, the quasidegenerate HOMO:LUMO designations from the simple Hückel-type model in [22] are subject to a strong Jahn–Teller distortion [88,89,90,91], which might lower the symmetry of the fullerene cage and introduces considerable electronic energy stabilization.

It is important to highlight that DFTB, like DFT, may provide a poor description of isomers with a quasidegenerate ground state, i.e., states with a pronounced multi-reference character for which a single Slater determinant is a bad approximation to the wave function. Various interesting and unexpected methodological problems might manifest themselves in this context [92,93,94,95]. Fortunately, such strongly correlated states do not occur very often; the study of all isomers of the classical fullerenes C_20_–C_50_ by Fowler, Mitchell, and Zerbetto [96] showed that only two out of 812 isomers (36:15 and 44:37) experience pronounced energy stabilization (>15 kcal/mol) in the approximate CISD calculations with four frontier orbitals, which suggests a pronounced multi-reference character of the underlying wave functions. DFTB and DFT would overlook such strongly correlated states, both predicting too high energies; for example, among the isomers of C_36_, DFTB predicts the isomers 36:14 and 36:15 within 0.5 kcal/mol (for an almost analogous DFT result, see [97]), but misses the fact that 36:15 is the ground state. This problem is rather serious, as there is no well-developed computational protocol for establishing univocally reliable energy rankings of fullerene isomers at their optimized geometries [98]. An obvious candidate for computing such an energy ranking of fullerene isomers would be the CASSCF/PT2 scheme (see for example [99]); however, creating such a ranking would constitute a considerable computational effort, and has not been yet performed in a systematic manner.

Note that these issues, while methodologically interesting and definitely worth further investigation in future studies, do not pose any serious problems for our current plans; Sure et al. [22] reported that the triplet and quintet isomers of C_60_ have relatively high energies, with the lowest (60:1728) lying 116.1 kcal mol^−1^ above the energy of the most stable isomer and the highest (60:44) lying 344.7 kcal mol^−1^ above this energy. These separations could perhaps be somewhat reduced by the Jahn–Teller distortions, but we do not expect this effect to be large. In our DFTB study, both the 60:1728 and 60:44 isomers possess a clear closed-shell singlet character with considerable HOMO:LUMO gaps (5.41 and 11.0 eV, respectively), while their stability is not altered significantly (their DFTB energies are respectively 112.3 and 333.2 kcal mol^−1^ above the DFTB energy of the most stable isomer). While we discuss this problem here in detail in order to inform readers about the possible difficulty, we do not think that the fraction of the remaining open-shell isomers (49 out of 29,767 cases, representing 0.16% of the total number of studied isomers) could statistically alter the results of the current analysis. Most (36 out of 49) of the relative energies of these open-shell structures are larger than 100 kcal mol^−1^, and for only four isomers (58:1205, 52:425, 68:6081, and 58:1151) is the relative energy smaller than 75 kcal mol^−1^. The most serious interpretational difficulty remains for the 58:1205 isomer, which has the lowest energy out of all isomers of C_58_. Again, this is rather fortunate for our analysis, as more rigorous quantum chemical calculations with a definitive value of the spin quantum number can only lower this value, and cannot alter the fact that 58:1205 corresponds to the most stable isomer of C_58_.

The topological invariants are computed in the form of the ZZ polynomial [100,101] for each isomer. Brief graph-theoretical definitions of the underlying concepts are provided below; for further details and explanations, the reader is referred to the rich existing literature on this topic [73,102,103,104,105,106,107,108]. From a graph-theoretical point of view, a fullerene isomer can be expressed as a 2-connected finite-plane graph **B**, with twelve interior faces being pentagons and n/2−10 interior faces corresponding to hexagons. Such a graph is usually represented by the corresponding Schlegel diagram; for examples of Schlegel diagrams, see Figure 2. A Kekulé structure **K** is defined as a spanning subgraph of **B** of which all components are isomorphic to a complete graph on two vertices ($K_{2}$). The number of distinct Kekulé structures **K** that can be constructed for **B** is referred to as the Kekulé count K. Similarly, a Clar cover **C** is defined as a spanning subgraph of **B** of which all components are isomorphic to either $K_{2}$ or to a cycle of girth six ($C_{6}$). The number of distinct Clar covers **C** that can be constructed for **B** is referred to as the Clar count C. Note that in the chemical literature one usually refers to $K_{2}$ as a double bond and to $C_{6}$ as an aromatic Clar sextet; similarly, a Kekulé structure **K** is referred to as a resonance structure of **B** that can be constructed using n/2 double bonds and a Clar structure **C** is referred to as a generalized resonance structure of **B** that can be constructed using *k* aromatic Clar sextets and (n−6k)/2 double bonds. The maximal number of aromatic Clar sextets $C_{6}$ that can be accommodated in **C** is referred to as the Clar number Cl of **B**. The Clar covers with Cl aromatic sextets $C_{6}$ are referred to as the Clar formulas of **B**. The Clar covers with *k* aromatic sextets $C_{6}$ are referred to as the Clar covers of order *k*. If we represent the number of Clar covers of order *k* for **B** by $c_{k}$, we can define a combinatorial polynomial
(1)$$\mathrm{ZZ}\left(B,x\right)=\sum_{k=0}^{Cl}c_{k}\;x^{k},$$
usually referred to in the literature as the Clar covering polynomial, Zhang–Zhang polynomial, or in short the ZZ polynomial of **B**. Clearly, the ZZ polynomial of **B** has the following properties:The number of Kekulé structures of **B** is provided by $K=c_{0}=\mathrm{ZZ}\left(B,0\right)$.The number of Clar covers of **B** is provided by $C=c_{0}+\cdots +c_{Cl}=\mathrm{ZZ}\left(B,1\right)$.The Clar number of **B** is provided by $Cl=\deg\left(\mathrm{ZZ}\left(B,x\right)\right)$.The number of Clar formulas of **B** is provided by $c_{Cl}=\mathrm{coeff}\;\left(\mathrm{ZZ}\left(B,x\right),x^{Cl}\right)$.The ZZ polynomial $\mathrm{ZZ}\left(B,x\right)$ is a generating function for the sequence $\left[c_{0},c_{1},\ldots ,c_{Cl}\right]$ of Clar covers of different orders.

These theoretical concepts are illustrated here using the simple example of corannulene (C_20_H_10_), a non-planar molecule with a close relation to IPR (5,6)-fullerenes. All 31 Clar covers of corannulene are shown in Figure 3. The Clar covers are grouped in three classes: $\left(i\right)$ eleven Kekulé structures, i.e., Clar covers of order 0; $\left(ii\right)$ fifteen Clar covers of order 1; and $\left(iii\right)$ five Clar formulas, i.e., Clar covers of maximal conceivable order Cl (here, Cl=2). Consequently, the ZZ polynomial of **B** = corannulene is provided by $\mathrm{ZZ}\left(B,x\right)=11+15x+5x^{2}$, the Kekulé count is K=11, the Clar count is C=31, and the Clar number of corannulene is Cl=2. This example also illustrates that determining topological invariants of fullerenes can be quite a cumbersome problem. Fortunately, ZZ polynomials can be conveniently and readily computed owing to their recursive properties related to the molecular graph decomposition tree (for more information, see Properties 1–7 in [103]). Consequently, the ZZ polynomial of an arbitrary benzenoid or fullerene **B** can be efficiently computed using recursive decomposition algorithms [102,103,105] or readily determined using the interface theory of benzenoids [108,109,110,111,112,113]. A useful practical tool for determining the ZZ polynomials of general planar benzenoids is ZZDecomposer [105,106], which allows the user to define the related molecular graph using a provided graphical interface and perform all computations in an automatic fashion. For fullerenes, it is probably more convenient to use ZZPolyCalc instead [114], which reads the molecular geometry XYZ file or the adjacency matrix as an input. ZZPolyCalc is also considerably faster owing to an efficient algorithm involving intermediate fragments caching. Both programs are freely downloadable [115,116,117] and self-explanatory. In the current study, we have used the ZZPolyCalc software for all the topological invariant calculations. The time needed to compute a single ZZ polynomial is less than one tenth of a second for the studied here fullerenes.

The molecular XYZ structures of fullerene isomers have been generated in the following way. The ring-spiral pentagon lists for the fullerene isomers have been taken from the House of Graphs fullerene database [1]. The spiral sequences served as an input to the program FULLERENE [118] (Version 4.5), and the initial geometries of the isomers were generated using the adjacency matrix eigenvector (AME) method [3]. These structures were subsequently optimized using a force-field approach [119], with additional extension to account for the third bond type and additional dihedral angles (activated using the ‘iopt=3’ flag in the FULLERENE program). In instances when this procedure failed to generate a meaningful converged geometry, the process was repeated using Tutte embedding (3D-TE) [120] instead of AME and optimized using the ‘iopt=2’ flag in the FULLERENE code. The force field-preoptimized molecular structures of the C_52_–C_70_ fullerene isomers were subsequently optimized using a series of DFTB geometry optimizations, in which the atomic forces convergence criterion was gradually tightened from $10^{-2}$ to $10^{-7}$ in an alternating sequence of conjugate gradient/steepest descent optimization steps.

## 3. Results

The abundant amount of data generated in the current study prevents us from presenting it directly within the body of the paper. Most of the resulting data are presented in the Appendix A accompanying this study; here, we only analyze the most important features of the results. The file ZZpolynomials.txt lists the computed ZZ polynomials for all the fullerene isomers C_20_–C_70_, with the data for fullerenes C_52_–C_70_ computed in the current work and the data for fullerenes C_20_–C_50_, listed again here for the convenience of the reader, taken from our previous work [73]. The file Correlations.txt lists the topological invariants (Clar count C, Kekulé count K, and Clar number Cl) of all the fullerene isomers C_20_–C_70_ together with their optimized DFTB energies. The file DFTBresults.txt briefly summarizes our DFTB calculations, providing the total DFTB energy *E*, Mermin free energy $E_{\mathrm{Mer}}$, final gradient value of the optimized structure, and labels of the highest doubly-occupied (HOMO) and lowest unoccupied (LUMO) DFTB orbitals along with their orbital energies $\varepsilon_{\mathrm{HOMO}}$ and $\varepsilon_{\mathrm{LUMO}}$. In cases where it was not possible to converge DFTB to a closed-shell solution, an additional notification “HOMO-LUMO 0 gap”, along with the occupation pattern of the degenerate frontier orbitals, is added the the end of the pertinent line to inform the reader that a finite electronic temperature (T=0.00001) has been used to smear out the orbital occupation numbers in the vicinity of the Fermi energy. Note that while the total energy *E* and Mermin free energy $E_{\mathrm{Mer}}$ are different in these cases, the numerical difference between these two quantities is too small to have any sizable effect on our conclusions. The XYZ files containing the optimized DFTB geometries for each of the fullerene isomers can be found in the file Geometries.tar.xz.

### 3.1. Correlation of the Kekulé Count and Clar Count with the Total Energies of Fullerene Isomers

The correlations of the total DFTB energies *E* with the corresponding values of the Kekulé count K and the Clar count C for all the isomers of C_68_ and C_70_ are shown in Figure 4. Similar plots for smaller fullerenes are presented in the Appendix A; the resulting correlations resemble those shown in Figure 4. Two families of plots are obtained: one presenting the correlation between the Kekulé count K and the total energy *E*, and the second presenting the correlation between the Clar count C and total energy *E*. The distinction between isomers with different values of Cl is achieved by using the symbol scheme explained in the legend of Figure 4.

The most important observation is that the topological invariants C and K do not correlate strongly with the total DFTB energy of fullerene isomers; the shapes of the $E=E\left(C\right)$ distributions are irregular and somewhat ellipsoidal, and show no correlation whatsoever, while the shapes of the $E=E\left(K\right)$ distributions rather surprisingly show weak anti-correlation. The isomers with a large value of Kekulé count are usually the highest in energy, which contradicts the usual organic chemistry rule of thumb stating that the structural isomers with the largest number of resonance structures are the most stable. Apparently, curved fullerenes do not adhere to this rule, and seem to promote an opposite principle, that is, that the isomers with large K are the least stable. This observation is not really new; as described in Section 1, a similar conclusion was already drawn for C_60_ more than 25 years ago [30,35,36,71]. The anti-Kekulé principle seems to prevail here; for six out of ten fullerenes studied here (C_54_, C_56_, C_60_, C_62_, C_68_, and C_70_) the isomer with the largest number of Kekulé structures is the highest in energy, while for two further fullerenes (C_52_ and C_64_) the highest-energy isomer has a value of K very close to the maximal one.

### 3.2. Correlation between the Clar Number and Total Energy of Fullerene Isomers

The anti-Kekulé correlation mentioned in the previous paragraph seems to suggest that fullerenes prefer structures in which the hexagons assume an aromatic benzene-like geometry without the double–single bond alternation characteristic for Kekulé structures. This observation is indeed confirmed by the observation that the most thermodynamically stable isomers usually have the largest value of Cl; readers should recollect that the Clar number Cl indicates the largest number of aromatic Clar sextets that can be simultaneously accommodated inside a given isomer without violating chemical bonding principles. In Table 1, we present the populations of isomers of the fullerenes C_52_–C_70_ with a given value of Cl. The group with the most thermodynamically stable isomer is underlined. For six out of ten fullerenes (C_56_, C_60_, C_62_, C_64_, C_68_, and C_70_) the most thermodynamically stable structure belongs to the group with maximal value of Cl, while for three further fullerenes (C_52_, C_54_, and C_66_) it belongs to the group with the second largest value of Cl. It would be interesting to extend our study to larger fullerene cages in order to test the hypothesis that the most stable isomer of large fullerenes C_*n*_ with n>70 always maximizes the number Cl of aromatic Clar sextets.

The hypothesis stating that isomers with the largest conceivable value of Cl for a given fullerene correspond to the most thermodynamically stable structure might have important practical consequences should it be confirmed for fullerenes larger than C_70_. At the moment, finding the most energetically stable structural isomer of large fullerenes is relatively complex due to the quite substantial computational resources required to accomplish this task. Two factors contribute to the cost here: $\left(i\right)$ geometry optimization for larger fullerene cages C_*n*_ becomes more and more costly with growing *n*, and $\left(ii\right)$ a large number of isomers exists for large *n* that need to be screened in the search process. The number of isomers of C_*n*_ grows with *n*, such as $n^{9}$ [121,122]; for C_70_, it is 8,149, while for C_100_ it is already 285,913. Selecting only those isomers with the maximal conceivable Cl allows the group of candidates for the lowest-energy structure to be reduced considerably. For example, for C_70_, the number of isomers with maximal Cl=9 is 267, which accounts for 3.3% of the total number of isomers, while for C_100_ the number of isomers with maximal Cl=9 is 1442, which accounts only for 0.5% of the total number of isomers. Thus, confirming that the most stable isomer of large fullerenes C_*n*_ always maximizes the number Cl of aromatic Clar sextets could lead to considerable savings during identification.

### 3.3. Correlation between Clar and Kekulé Counts and Relation to Isomer Stability

The Clar and Kekulé counts are not fully independent. The correlation coefficient $R^{2}$ between these two measures varies from 0.49 to 0.59 depending on the size of the fullerene. Additionally, their relationship exhibits very interesting and distinct patterns. In Figure 5, we present a graph showing the Clar counts C as a function of Kekulé counts K for all isomers of $C_{68}$ and $C_{70}$. Analogous graphs for the remaining fullerenes $C_{52}$–$C_{66}$ are available in the Appendix A. The relationship between Clar and Kekulé counts in all cases has the shape of a slanted wedge. The approximately linear lower boundary has a slope that tends to increase with the system size (C/K > 3.87 for $C_{52}$ and C/K > 8.07 for $C_{70}$). The upper boundary is less regular, but generally tends to change with approximately the fourth power of K.

While the consistent shape of the Cvs.K distributions is quite interesting, the most surprising aspect of these graphs lies elsewhere; in almost all of the studied cases, the 30 most stable isomers of each fullerene are located almost entirely at the upper boundary of the wedge, with the most stable structure having the largest C/K ratio for $C_{60}$, $C_{66}$, and $C_{70}$. This regularity is quite visible in Figure 5 and Appendix A, where the 30 most stable isomers are depicted in different colors, with the most stable isomer represented by a red circle. For certain fullerenes, particularly the smaller ones, the detected pattern is weaker; for example, for $C_{58}$ the most stable structure is not near the upper boundary of the distribution, and several top-30 isomers are actually closer to the lower boundary of the distribution. Nevertheless, the discovered pattern of stable isomers grouping near the upper boundary becomes more pronounced as the size of the fullerene increases, and as such can become a very useful tool in discriminating the most stable isomers of larger fullerenes. We believe that this aspect of our study deserves further investigation.

### 3.4. Does the Most Stable Isomer Maximize the Kekulé Count among the Isomers with the Maximal Value of Clar Number?

Let us now verify the main hypothesis of the current work. In 2010, Zhang, Ye, and Liu [71] made the observation that for C_60_ the most energetically stable structural isomer maximizes the number of Kekulé structures K among the isomers with the maximal conceivable value of Cl. The question we would like to test here is whether such an observation is also correct for other fullerenes. The observation made by Zhang et al. [71] is clear from the current work, as is obvious from the right panel of Appendix A. There are eighteen isomers of C_60_ with the maximal value of Cl=8, and the lowest in energy, the icosahedral structure 60:1812, indeed maximizes the value of K among them. Is a similar observation true for the other fullerenes studied here? The answer is negative; C_60_ is the only fullerene among C_52_–C_70_ for which the observation made by Zhang et al. [71] is valid. Interestingly, an anti-Kekulé-like rule works much better here: for four of the studied fullerenes (C_56_, C_62_, C_68_, and C_70_), the isomer with maximal K among isomers with maximal Cl is the highest in energy! For the remaining five fullerenes (C_52_, C_54_, C_58_, C_64_, and C_66_), the isomer with maximal K among isomers with maximal Cl has intermediate energy; none of these isomers is a good candidate for a global energy minimum. Therefore, the observation of Zhang, Ye, and Liu for C_60_ is not useful for other fullerenes, and we can refute K as a useful indicator of fullerene stability in general. Note that the results presented previously in [73] for C_20_–C_50_ provide further support to this conclusion.

It seems, however, that the observation of Zhang, Ye, and Liu [71] for C_60_ can be made somewhat more useful if we introduce a small correction. Specifically, we shall now test a new hypothesis stating that the most stable isomer is the one which maximizes the Clar count C among the isomers with the maximal conceivable value of Cl. This observation is clearly correct for C_60_; the icosahedral isomer not only maximizes C among the isomers with Cl=8, it also maximizes C in the whole population of C_60_ isomers. A similar observation is true for two other fullerenes, C_66_ and C_70_, while for C_64_ the three isomers with maximal C correspond to the second, third, and fourth most stable isomers of this fullerene. Unfortunately, for the other fullerenes an isomer that maximizes C among the isomers maximizing Cl corresponds either to an intermediate energy structure (for C_52_, C_54_, and C_58_) or to the highest energy structure (C_56_, C_62_, and C_68_). The last sequence with the progression n→n+6 is quite interesting, suggesting that the isomers of $C_{6m+2}$ with m≥9 that maximize C could correspond to the structural isomers with the highest energy. This hypothesis should be tested when the data for larger fullerenes become available. Summarizing the observations in this paragraph, we can state that the Clar count C is only marginally more useful than K as a topological indicator for characterizing fullerene stability.

### 3.5. ZZ Polynomials Can Be Used as Alternative Unique Labels for Discriminating between Fullerene Isomers

Another interesting and possibly useful observation, though unrelated to the energetic stability of fullerenes, is the fact that all 29,767 of the ZZ polynomials computed in the current work are distinct from each other and can be used to discriminate between different isomers of fullerenes C_52_–C_70_. This observation also extends to the 812 different isomers of C_20_–C_50_ studied by us previously. Thus, all 30,579 isomers of fullerenes C_20_–C_70_ have different ZZ polynomials! Initial checks suggest that this uniqueness of ZZ polynomials extends to larger fullerene cages as well. Consequently, ZZ polynomials might be used to label different isomers of fullerenes as an alternative to the canonical spiral sequence. Calculation of ZZ polynomials can be performed almost instantaneously for the fullerenes studied here, providing a very convenient method of recognizing which particular fullerene isomer is currently considered. Note that determination of the ZZ polynomial can be performed directly from the XYZ geometry file of a given isomer or from its topological adjacency matrix, as ZZ polynomials are invariant with respect to vertex permutation and with respect to geometrical transformation (rotations, translations, and deformations) of the fullerene cage. This particular property of ZZ polynomials makes them a very convenient descriptor for machine learning models, providing a meaningfully unique multilabel for each isomer that consists of its various topological invariants.

### 3.6. Pauling Bond Orders in Fullerenes

Access to all possible Kekulé structures and all possible Clar covers of fullerene isomers allows us to compute the Kekulé and Clar bond orders in the way first introduced by Pauling [123,124] and popularized by Herndon, Randić, and others in the 1970s [125,126,127,128,129,130,131,132,133]. Our study can be considered as a direct extension of the work on Pauling–Kekulé bond orders carried out by Narita, Morikawa, and Shibuya for the most stable isomers of C_60_ and C_70_ [134] and the work on Pauling–Clar bond orders by Randić for C_60_ [135]. Recent years have provided evidence of multiple situations in which Kekulé structure-based models yield evidently incorrect predictions [136,137,138,139,140]. Here, we use the current results on a very large statistical sample to verify whether the the Pauling–Clar bond orders and Pauling–Kekulé bond orders, both of which are quantities computable directly from the topology of bond connections without any use of quantum chemical theory, can be of any value for practical purposes in the theory of fullerene isomers.

The procedure is simple. First, we choose a particular bond *B* in a given fullerene isomer and inspect the π character assigned to it by each Kekulé structure or Clar cover. For Kekulé structures, there are only two possibilities: *B* can be a single bond (with the π bond order of 0) or a double bond (with the π bond order of 1). For Clar covers, there are three possibilities: *B* can be a single bond (with the π bond order of 0), a double bond (with the π bond order of 1), or a member of an aromatic Clar sextet (with the π bond order of $\frac{1}{2}$). An average of all these contributions over the full set of Kekulé structures produces the classical Pauling bond orders for fullerenes [123,134]. A similar average computed over the full set of Clar covers produces the modified Pauling–Clar bond orders [135]. The easiest way to estimate the usefulness of these quantities is to correlate them with the bond lengths obtained via DFTB optimization of each structure. In this way, each bond can be represented as a dot, with the *x*-coordinate corresponding to the bond order and the *y*-coordinate corresponding to the bond length in the DFTB-optimized structure of a given isomer. Such correlations for all 2,978,872 bonds in all 30,579 isomers of fullerenes C_20_–C_70_ are shown in Figure 6, with the green dots representing the Pauling–Clar bond orders and the purple squares representing the Pauling–Kekulé bond orders. For reasons of technical feasibility, only 1% of the randomly selected points out of 2,978,872 are shown in Figure 6; notably, this limitation does not alter the visual distribution of the points. Both distributions show clear correlations between the bond orders and bond lengths; single bonds are longer and double bonds are shorter, which is in close agreement with chemical intuition. The correlations are rather modest, with $R^{2}$ coefficients of 0.486 and 0.429 for the Pauling–Clar and Pauling–Kekulé bond orders, respectively. It is clear that the Pauling–Clar bond orders are more useful in practice thanks to providing better discrimination between shorter and longer bonds. The $R^{2}$ coefficient of 0.486 shows that approximately half of the statistical variance is explained by the predicted linear trend plotted in Figure 6, while the other half cannot be inferred from the bond order alone. This regularity can be very useful for solving one of the main problems associated with generation of XYZ geometry files for fullerene isomers, where the preparation of a physically relevant initial geometrical structure from the adjacency matrix computed on the basis of the spiral sequence is a challenging task. Because the bond orders can be computed directly from the adjacency matrix and correlate with the bond lengths, it is possible to prepare a good initial geometry file directly on the basis of the bond lengths inferred from the bond orders.

## 4. Conclusions

We report a compilation of topological invariants for all 29,767 structural isomers of the carbon (5,6)-fullerenes C_52_–C_70_. The results are presented in the file ZZpolynomials.txt in the Appendix A. This collection of data, together with the previously reported [73] ZZ polynomials for fullerenes C_20_–C_50_, completes our determination of the most important topological invariants for (5,6)-fullerenes with 70 or fewer carbon atoms. Interestingly, all of the ZZ polynomials computed for the 30,579 isomers of fullerenes C_20_–C_70_ are distinct, and this uniqueness seems to extend to larger fullerene cages as well, making the ZZ polynomials a convenient label for identifying and discriminating between various fullerene isomers with potential applications to machine learning models.

The computed Clar numbers, Clar counts, and Kekulé counts of the C_52_–C_70_ isomers are correlated with the total DFTB electronic energies computed at the optimized DFTB geometries of the corresponding fullerene cages (the DFTB energies are listed in the file DFTBresults.txt, and the optimized DFTB geometries are provided in the file Geometries.tar.xz, both of which accompany this paper in the Appendix A). The correlations are computed in order to verify the hypothesis of Zhang, Ye, and Liu [71], who postulated that the most energetically stable structural isomer of C_*n*_ maximizes the Kekulé count K among the isomers with the maximal conceivable Clar number Cl. Analysis of our data shows that this hypothesis only holds for C_60_. For the remaining nine fullerenes (C_52_–C_58_ and C_62_–C_70_), the isomers with maximal K among the isomers with maximal Cl correspond to high or intermediate DFTB energies; none of these isomers are a good candidate for a global energy minimum of C_*n*_. Note that the results presented previously in [73] for C_20_–C_50_ provide further support to this conclusion. In general, our results suggest that both Kekulé count and Clar count are rather poor descriptors and predictors of isomer stability, while the Clar number, i.e., the maximal number of aromatic sextets, correlates better with the stability of isomers; however, its practical usefulness is limited. The most promising feature of our Clar and Kekulé count analysis is the observation that for larger fullerenes the most stable isomers are almost entirely located at the upper boundary of the CvsK distributions (for details, see Figure 5 and Appendix A). This observation can be very useful for prescreening isomers of larger fullerenes in order to identify viable candidates for their ground state.

Access to the complete sets of Kekulé structures and Clar covers allows us to compute the Pauling bond orders for fullerene isomers, which can be compared to the bond lengths obtained from quantum chemical calculations. The resulting Pauling–Clar and Pauling–Kekulé bond orders show rather modest correlations with the bond lengths, with the $R^{2}$ coefficients of 0.486 and 0.429, respectively. The Pauling–Clar bond orders are slightly more useful in practice, having better predictive power, and for example can be used in initial optimization of topology-generated fullerene cages.

An interesting aspect of our work is the discovery that a significant number of fullerene isomers that are predicted by a simple Hückel model to have open-shell electronic character experience a pronounced Jahn–Teller distortion, which leads to a transition to a lower-energy closed-shell state. While this should not be surprising in the light of the earlier results reported by Paulus [141] for C_20_–C_36_, final confirmation of these results may require additional CASSCF calculations in order to avoid artificial Jahn–Teller-like effects [92,93]. Notably, this pattern is observed in almost all the open-shell isomers of C_60_ studied in [22]. This finding implies that the conclusions of [22] regarding the abundance and stability of such isomers might require reevaluation to incorporate the influence of these Jahn–Teller effects.

## Figures and Tables

**Figure 1 molecules-29-04013-f001:**
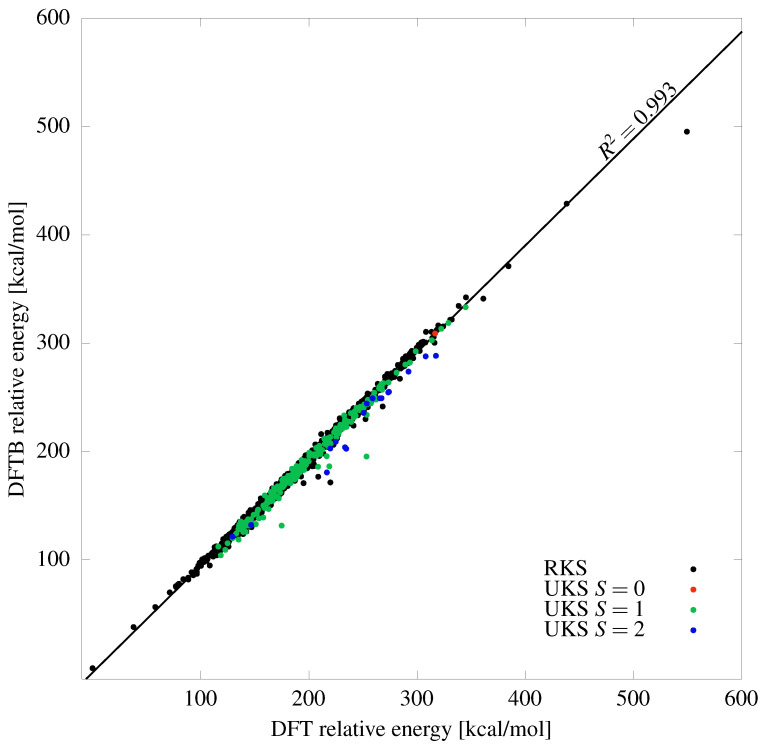
Comparison of the optimized DFTB and DFT energies for the 1812 isomers of C_60_, showing good correlation between these two methods. The color designation of points is described in the legend, and corresponds to the DFT calculations from [22].

**Figure 2 molecules-29-04013-f002:**
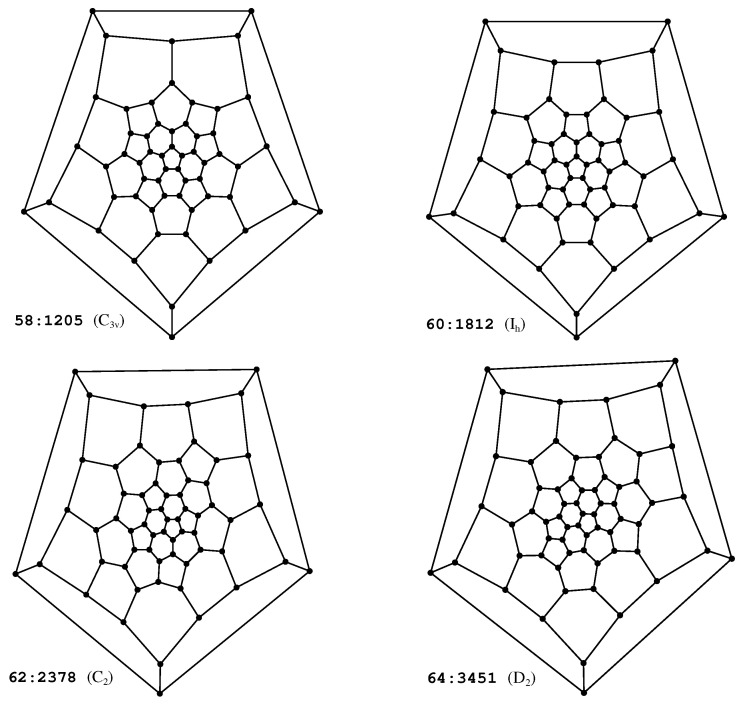
Schlegel diagrams of the most stable isomers of C_58_ (**upper left**) and C_60_ (**upper right**), clearly showing the close structural resemblance between these two isomers. For the most stable isomers of C_62_ (**lower left**) and C_64_ (**lower right**), the structural similarity with C_60_ is less pronounced.

**Figure 3 molecules-29-04013-f003:**
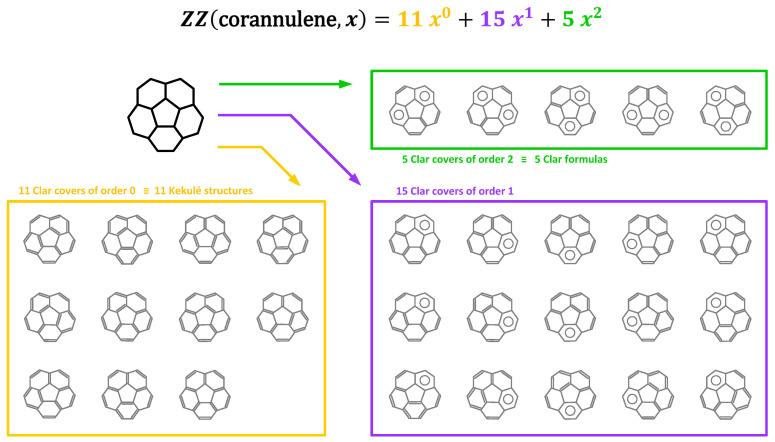
Clar covers (i.e., extended resonance structures) of corannulene (C_20_H_10_) can be conveniently enumerated using the ZZ polynomials, which keep track of the number of resonance structures for each order. The order *k* of each Clar cover **C** is defined as its number of aromatic Clar sextets.

**Figure 4 molecules-29-04013-f004:**
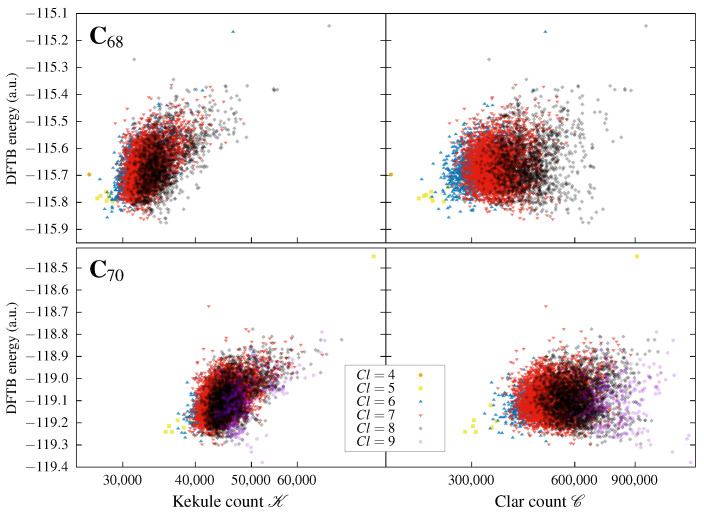
Total DFTB energies of all isomers of $C_{68}$ (upper panels) and $C_{70}$ (lower panels) plotted as a function of their topological invariants, i.e., the Kekulé count K (left panels) and Clar count C (right panels). The information about the Clar number Cl is conveyed via the symbol code explained in the legend. The plots show weak anti-correlation between *E* and K and no correlation between *E* and C. Similar tendencies are observed also for smaller fullerenes C_52_–C_66_. For details, see the Appendix A.

**Figure 5 molecules-29-04013-f005:**
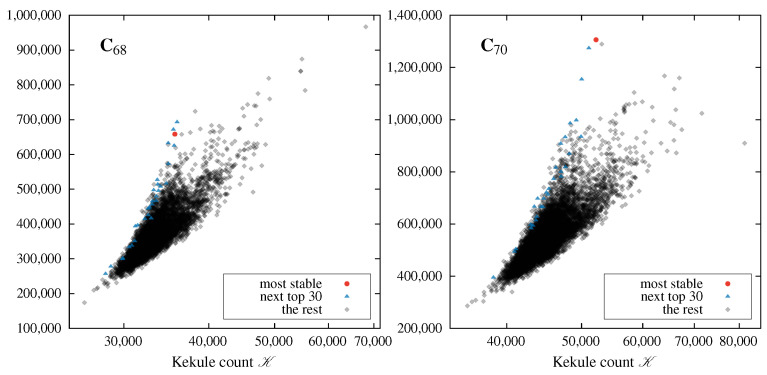
Clar counts C as a function of Kekulé counts K for $C_{68}$ (left panel) and $C_{70}$ (right panel). The most stable structure is marked with a red dot while structures 2–30 (ordered by stability) are marked with blue triangles. The most stable isomers tend to group near the top leftmost part of the graph.

**Figure 6 molecules-29-04013-f006:**
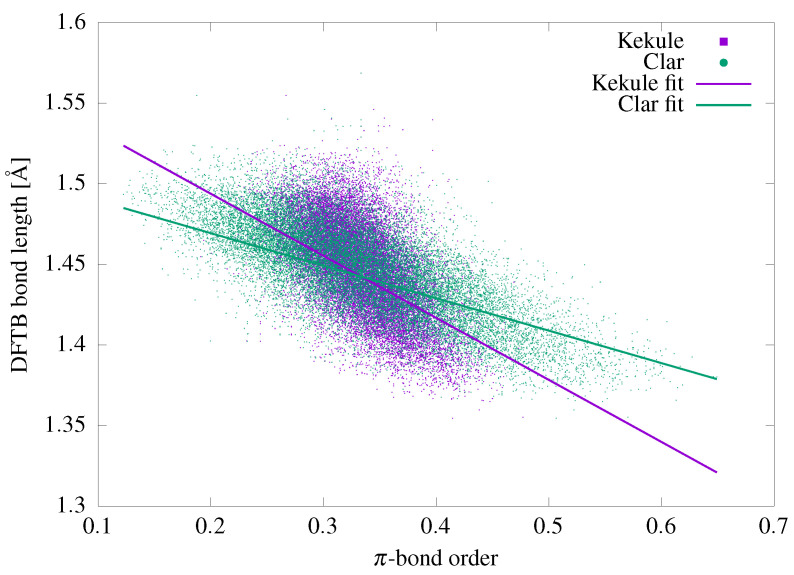
Correlation between the Kekulé and Clar bond orders and corresponding DFTB bond lengths for all isomers of the C_20_–C_70_ fullerenes. The linear fits are 1.5096−0.2016x and 1.5708−0.3851x for the Clar and Kekulé models, respectively. For clarity, the graph displays data for only 1% of the total number of 2,978,872 bonds (chosen randomly). The fits have been performed on the complete sets.

**Table 1 molecules-29-04013-t001:** The number of isomers of fullerenes C_52_–C_70_ with a given value of Clar number Cl. The relative abundance of each group is provided in parentheses. The group containing the most stable isomer is underlined. For six fullerenes (C_56_, C_60_, C_62_, C_64_, C_68_, and C_70_), the most thermodynamically stable structure belongs to the group with maximal value of Cl; for three further fullerenes (C_52_, C_54_, and C_66_), it belongs to the group with the second largest Cl. The presented data suggest that for larger fullerenes the most thermodynamically stable isomer can be found by studying only the isomers with large values of Cl.

Fullerene	Cl					
	4	5	6	7	8	9
C_52_	116 (27%)	254 (58%)	67 (15%)			
C_54_	35 ( 6%)	452 (78%)	86 (15%)	7 ( 1%)		
C_56_	32 ( 3%)	453 (49%)	439 (48%)			
C_58_	2 ( 0%)	506 (42%)	597 (50%)	100 ( 8%)		
C_60_	6 ( 0%)	290 (16%)	1316 (73%)	182 (10%)	18 ( 1%)	
C_62_	1 ( 0%)	198 ( 8%)	1468 (62%)	718 (30%)		
C_64_		53 ( 1%)	1937 (56%)	1280 (37%)	195 ( 6%)	
C_66_		33 ( 1%)	1342 (30%)	2817 (63%)	275 ( 6%)	11 (0%)
C_68_	1 ( 0%)	8 ( 0%)	1109 (18%)	3806 (60%)	1408 (22%)	
C_70_		8 ( 0%)	412 ( 5%)	5186 (64%)	2276 (28%)	267 (3%)

## Data Availability

Most of the research results, including the optimized DFTB energies and geometries of all the isomers of the fullerenes C_20_–$C_{70}$ and their corresponding ZZ polynomials, can be downloaded at: https://www.mdpi.com/article/10.3390/molecules29174013/s1 as the Appendix A for this paper. For additional material, the reader is encouraged to contact the authors.

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
