# Peer review of "Kekulé Counts, Clar Numbers, and ZZ Polynomials for All Isomers of (5,6)-Fullerenes C_52_–C_70"

_molecules, 2024, doi:10.3390/molecules29174013_

Round 1

Reviewer 1 Report

Comments and Suggestions for Authors

This is a fine well executed study of topological indices of fullerene stability. The results largely refute common hypotheses, but this in itself is a valuable outcome, which is worth publishing. 

Main remarks:

- The authors have now accumulated a large set of Kekule and Clar counts for a vast amount of fullerene isomers. The question which then comes to mind is to what extent these two indices are really independent, and here is the opportunity to check their correlation. This may be some extra work, but I think it is worth pursuing.

- The literature is well covered, except perhaps for the studies by Manuel Alcami and Fernando Martin on non IPR isomers. In their papers focus is on charge distribution in the cage, but this is also related to the open shell theme which pops up in the present context. I would appreciate some comments here. Open-shell structures might indeed be susceptible to charge separations. Are these open-shell structures mostly non-IPR ?

Comments on the Quality of English Language

English language is fine, but I noted a few errors:

- p. 1, line 34: and some not attracted --> while some did not attract

- p. 4, line 160: singly degenerate --> non-degenerate

- p. 10, line 373: progression --> sequence

- p. 12, line 402: an direct --> a direct

Author Response

Comment 1:    The authors have now accumulated a large set of Kekule and Clar counts for a vast amount of fullerene isomers. The question which then comes to mind is to what extent these two indices are really independent, and here is the opportunity to check their correlation. This may be some extra work, but I think it is worth pursuing.

Response 1:    We would like to thank the referee  for this suggestion. The Kekule and Clar counts are indeed somewhat correlated and form some distinct patterns. Plotting Clar counts vs Kekule counts helped us to identify some qualitative trends that may help to identify the most stable isomers. The results of the analysis are probably best described by quoting verbatim the new Section 3.3 which has been added to our manuscript to follow the suggestion of the referee:

"3.3. Correlation between Clar and Kekulé counts and its relation to the isomers stability

The Clar and Kekulé counts are not fully independent. The correlation coefficient R2 between these two measures varies from 0.49 to 0.59 depending on the size of the fullerene. Additionally, their relationship exhibits very interesting and distinct patterns. In Fig. 5, we present a graph showing the Clar counts C as a function of Kekulé counts K for all isomers of C68 and C70. Analogous graphs for the remaining fullerenes C52–C66 are available in the Supplementary Materials as Figure S3. The relationship between Clar and Kekulé counts in all cases has a shape of a slanted wedge, with approximately linear lower boundary, whose slope tends to increase with the system size (C/K > 3.87 for C52 and C/K > 8.07 for C70). The upper boundary is less regular but generally tend to change approximately with the fourth power of K. While the consistent shape of the C vs K distributions is quite interesting, the most surprising aspect of these graphs lies elsewhere. Namely, in almost all of the studied cases, the 30 most stable isomers of each fullerene are located almost entirely at the upper boundary of the wedge, with the most stable structure having the largest C/K ratio for C60, C66, and C70. This regularities are well visible in Figs. 5 and S3, where the 30 most stable isomers are depicted in color with the most stable isomer represented by a red circle. For some fullerenes, particularly the smaller ones, the detected pattern is weaker. For example, for C58 the most stable structure is not near the upper boundary of the distribution, and several top-30 isomers are actually closer to the lower boundary of the distribution. Nevertheless, the discovered pattern of stable isomers grouping near the upper boundary becomes more pronounced as the size of the fullerene increases, and as such can become a very useful tool in discriminating the most stable isomers of larger fullerenes. We believe that this aspect of our study deserves further studies."

We consider this discovery (motivated by the remark of the referee, for which we again thank very much) very important. For this reason, we have mentioned it explicitly in the Conclusion section as "The most promising feature of our Clar and Kekulé count analysis is the observation that for larger fullerenes, the most stable isomers almost entirely are located at the upper boundary of the C vs K distributions (for details, see Figs. 5 and S3). This observation can be very useful for prescreening isomers of larger fullerenes in order to identify viable candidates for their ground state." and in absract as "[...] with the only exception of the Clar count/Kekulé count ratio, which at the moment
seems to be the most important diagnostic discovered from our analysis."

Comment 2: The literature is well covered, except perhaps for the studies by Manuel Alcami and Fernando Martin on non IPR isomers. In their papers focus is on charge distribution in the cage, but this is also related to the open shell theme which pops up in the present context. I would appreciate some comments here. Open-shell structures might indeed be susceptible to charge separations. Are these open-shell structures mostly non-IPR ?

Response 2: Only two isomers in our study possess IPR character, the Ih isomer of C60 and the D5h isomer of C70. The rest of the isomers (i.e., 29765 isomers) have non-IPR character. However, only in 49 cases (i.e., in less than 0.2% of cases), the fullerene isomers display distinct open-shell character corresponding to an accidental quasidegeneracy of the frontier orbitals. This also signifies that in more than 99.8% of cases, the non-IPR isomers have a well-defined closed-shell character with a considerable HOMO-LUMO gap.  So accordingly, the answer to the referee's question is: Yes, the open-shell structures are non-IPR, but we do not believe that the non-IPR character of these structures is the main cause of the open-shell character, as more than 99.8% of the non-IPR isomers have closed-shell character. The charge distribution in the open- and closed-shell isomers is similar. Since DFTB automatically computes Mulliken charges (as one of its internal quantities), it is very easy to see the magnitude of the induced charges on each atom. These magnitudes are very small (i.e., each atom is almost neutral) and there are no noticeable differences between open- and closed-shell isomers.

The situation is  different in the case of anions of fullerenes, where the additional charge can be evenly/unevenly distributed among the carbon atoms, possibly giving natural, Coulomb-force based impulse for lifting high symmetry in those charged fullerene cages. However, even for these charged species, we believe that the main cause for the Jahn-Teller effect is the open-shell character of degenerate molecular orbitals, caused by the presence of the additional charge, rather than uneven distribution of the charge. 

We have examined the papers by Martin & Alcami in this context and found that most of them are not directly related to our study. However, we have discovered that two of these papers (J Nanosci. Nanotechnol. 7 (2007) 1329--1338 and J. Chem. Phys. 119 (2003) 5545--5557) in addition to yet another paper (Chem. Phys. Lett. 407 (205) 153--158), which has been already included in the original version of our manuscript, are supporting the discussion in the first Section, so the modified version of our manuscript contains 3 citations to Martin & Alcami work.

The referee's comment also made us realize that other citations would be helpful for the discussion of the Jahn-Teller effect in fullerenes. To this end we have added the three following citations:
1. Chancey, C.C.; O’Brien, M.C.M. The Jahn-Teller Effect in C60 and Other Icosahedral Complexes; Princeton University Press: Princeton, 1998.
2. Canton, S.E.; Yencha, A.J.; Kukk, E.; Bozek, J.D.; Lopes, M.C.A.; Snell, G.; Berrah, N. Experimental Evidence of a Dynamic Jahn-Teller Effect in C60+, Phys. Rev. Lett. 2002, 89, 045502. 
3. Liu, D.; Niwa, Y.; Iwahara, N.; Sato, T.; Chibotaru, L.F. Quadratic Jahn-Teller effect of fullerene anions. Phys. Rev. B 2018, 98, 035402.

Comment 3:     English language is fine, but I noted a few errors:

- p. 1, line 34: and some not attracted --> while some did not attract

- p. 4, line 160: singly degenerate --> non-degenerate

- p. 10, line 373: progression --> sequence

- p. 12, line 402: an direct --> a direct

Response 3:    We thank the referee for careful reading of our manuscript. The answers to the referee's suggestions are describes below.

- p. 1, line 34: and some not attracted --> while some did not attract

     The meaning of the original sentence "Characterization of the soot components and understanding the reasons for which some of the fullerene isomers are more abundant in the soot and some not, attracted considerable interest of the chemical community" was distorted by the missing comma sign (now added). Hence, the grammar that the referee was trying to correct in our sentence should be now correct and the meaning should be easy to understand.

- p. 4, line 160: singly degenerate --> non-degenerate

    Changed as suggested by the referee.

- p. 10, line 373: progression --> sequence

    The sentence "The last progression is..." is now changed to "The last sequence with the progression $n\!\rightarrow\! n\!+\!6 $ is...". This change reconciles the comment of the referee with our original intention, and it hopefully more clear to the reader.

- p. 12, line 402: an direct --> a direct

    Changed as suggested by the referee.

Reviewer 2 Report

Comments and Suggestions for Authors

Referee report

Molecules

Manuscript ID: molecules-3138415

Kekulé counts, Clar numbers, and ZZ polynomials for all isomers of (5,6)-fullerenes C52–C70

Authored by H.A. Witek, R. Podeszwa

The manuscript is devoted to the computation and analysis of topological invariants for all the isomers of carbon (5, 6)-fullerenes with n=52−70. These invariants, including Kekulé count, Clar count, and Clar number, are presented in the form of Zhang-Zhang (ZZ) polynomials. The study examines the distinctiveness of these ZZ polynomials for different isomer cages and explores their potential to uniquely identify various isomers. Additionally, the manuscript investigates the chemical applications of these invariants and their correlation with isomer stability and bond properties. It was found that: i) The ZZ polynomials are found to be unique for each isomer cage, providing a unique identifier for differentiating between various isomers; ii) There is a weak correlation between the topological invariants (Kekulé count, Clar count, Clar number) and the stability of the isomers. This finding questions the predictive power of these invariants in determining the most stable isomer of a given fullerene; iii) Stronger correlations are observed between Pauling bond orders, computed from Kekulé structures or Clar covers, and the equilibrium bond lengths obtained from optimized DFTB (Density Functional based Tight Binding) geometries for all the 30,579 isomers of C20-C70; iv) The correlation dependencies discovered in the study can be used in machine-learning procedures as descriptors. 

The conclusions highlight the significance of ZZ polynomials in the identification of fullerene isomers, while also questioning the traditional topological invariants' effectiveness in predicting isomer stability. The study emphasizes the importance of detailed computational approaches to understand fullerene chemistry. The topic of the manuscript is in the mainstream of modern science and technology, theoretical approaches used to discover the correlation dependencies properly match the goals of the study, the text of the manuscript is clearly written.

After careful examination of the manuscript, I have some minor remarks and criticisms: 

1.     p. 1: There is one exception for the expression of the number of hexagons n=22. ;

2.     p. 6: “These theoretical concept are illustrated...”  --> “These theoretical concepts are illustrated...”;

3.     p. 11, Figure 5 Legend; Green and purple dots should be properly described and purple linear fit should be added; 

4.     In the text authors speculated that the deviations of descriptors from linear fits are caused by “random reasons”. Actually, I cannot agree with such speculations since it is well known that some (if not many) fullerene-like atomic clusters display distinctive electronic correlations (see, for example, Lof R.W.,van Veenendaal M.A., Koopmans B., Jonkman H.T., Sawatzky G.A., Phys. Rev. Lett. 68, 3924 (1992); Knupfer M., Poirier D.M., Weaver J.H., Phys. Rev. B 49, 2281 (1994); B. Paulusa, Electronic and structural properties of the cage-like molecules C20 to C36, Phys. Chem. Chem. Phys. 5, 3364-3367 (2003)  https://doi.org/10.1039/B304539K; D. Stück, T.A. Baker, P. Zimmerman, W. Kurlancheek, M. Head-Gordon, On the nature of electron correlation in C60, J. Chem. Phys. 135, 194306 (2011) https://doi.org/10.1063/1.3661158; S.A. Varganov, P.V. Avramov, S.G. Ovchinnikov, M.S. Gordon, A study of the isomers of C36 fullerene using single and multireference MP2 perturbation theory, Chem. Phys. Lett., 362, 380-386 (2002).; S. Chakravarty, M. P. Gelfand, and S. Kivelson, Science 254, 970 (1991).; V. Ya. Krivnov, I. L. Shamovsky, E. E. Tornau, A. Rosengren, Electronic correlation effects in a fullerene molecule studied by the variational Monte Carlo method, Phys. Rev. B 50, 12144 (1994). https://doi.org/10.1103/PhysRevB.50.12144). Probably it could be better to somehow discuss the role of electronic correlations in determination of the key structural features and energetic characteristics of fullerene isomers.

The manuscript molecules-3138415 " Kekulé counts, Clar numbers, and ZZ polynomials for all isomers of (5,6)-fullerenes C52–C70" authored by H.A. Witek, R. Podeszwa can be published in Molecules after minor revision.

--------------

Disclaimer: All the works cited in this peer-reviewed report are cited only because of their critical importance to this review. The reviewer neither confirms or denies the authorship of all publications and does not require any of them to be cited in revised version of the manuscript.

Comments on the Quality of English Language

Some minor misprints should be corrected

Author Response

Comment 1.    p. 1: There is one exception for the expression of the number of hexagons n=22

Response 1.     We thank the referee for this comment. An appropriate information "except for $n\!=\!22$ for which no such isomer exists" has been included in the first paragraph

Comment 2.    p. 6: “These theoretical concept are illustrated...”  --> “These theoretical concepts are illustrated...”

Response 2:    Corrected as requested.

Comment 3.     p. 11, Figure 5 Legend; Green and purple dots should be properly described and purple linear fit should be added 

Response 3: The figure has been modified according to the referee's suggestions. The description has been given in the caption. We have also found that the previous version accidentally did not present all the possible bonds for these systems (data for C64-C70 was missing). We have updated the graph accordingly, and the values of the new linear fits are slightly different (the small  changes in the fitting coefficient values do not have serious consequences for the discussion in text).  

Comment 4. In the text authors speculated that the deviations of descriptors from linear fits are caused by “random reasons”. Actually, I cannot agree with such speculations since it is well known that some (if not many) fullerene-like atomic clusters display distinctive electronic correlations (see, for example, Lof R.W.,van Veenendaal M.A., Koopmans B., Jonkman H.T., Sawatzky G.A., Phys. Rev. Lett. 68, 3924 (1992); Knupfer M., Poirier D.M., Weaver J.H., Phys. Rev. B 49, 2281 (1994); B. Paulusa, Electronic and structural properties of the cage-like molecules C20 to C36, Phys. Chem. Chem. Phys. 5, 3364-3367 (2003)  https://doi.org/10.1039/B304539K; D. Stück, T.A. Baker, P. Zimmerman, W. Kurlancheek, M. Head-Gordon, On the nature of electron correlation in C60, J. Chem. Phys. 135, 194306 (2011) https://doi.org/10.1063/1.3661158; S.A. Varganov, P.V. Avramov, S.G. Ovchinnikov, M.S. Gordon, A study of the isomers of C36 fullerene using single and multireference MP2 perturbation theory, Chem. Phys. Lett., 362, 380-386 (2002).; S. Chakravarty, M. P. Gelfand, and S. Kivelson, Science 254, 970 (1991).; V. Ya. Krivnov, I. L. Shamovsky, E. E. Tornau, A. Rosengren, Electronic correlation effects in a fullerene molecule studied by the variational Monte Carlo method, Phys. Rev. B 50, 12144 (1994). https://doi.org/10.1103/PhysRevB.50.12144). Probably it could be better to somehow discuss the role of electronic correlations in determination of the key structural features and energetic characteristics of fullerene isomers.

Response 4.   We thank the referee for pointing out that fullerene systems may display strong electron correlation, and that single-determinant-based wave functions are often incapable of capturing this effect. For example, the Ih isomer of C60 (60:1812) shows a RHF/UHF instability, which, if interpreted as a Jahn-Teller distortion of the highly-symmetric Ih cage, might be an indication that open-shell, strongly-correlated, lower-symmetry and lower-energy structures of 60:1812 are more viable candidates for its ground state. This effect has been examined in detail (DOI 10.1039/C8CP07613H) and it has been found that this instability is artificial and can be attributed to the missing electron correlation in the HF description of 60:1812. On the other hand, C36 within the D6h point group (the isomer 36:15) is strongly correlated with a small energy difference between the triplet and singlet states. Similarly, C20 within the Ih point group is strongly correlated, but its Jahn-Teller distorted structures (C2hD2hCi, and D3h) have been found not to be strongly correlated and the underlying Hartree–Fock symmetry breaking is therefore artificial, being most likely caused by the missing electron correlation. Distinguishing between these two situations [real Jahn-Teller (JT) effect or artifical JT-like effect associated with the deficiencies of the used wave function ansatz] is quite difficult, as it requires using certain specialistic diagnostic measures (for details see DOI 10.1039/C8CP07613H) that are not always readily available for standard quantum chemical calculations. There are two important reasons suggesting that this difficulty should not concern our calculations to a large degree.

  1. Our calculation is based on the DFTB model, which can be considered as an approximation to DFT. For this reason, some portion of electronic correlation is already effectively included in the calculations and consequently our results should be less susceptible to the artificial JT-like effect due to missing electron correlation than the HF results discussed broadly in the literature quoted by the referee. Of course, the deficiencies of approximate DFT functionals (DFTB is based on the PBE parametrization of DFT) may always lead to situations when the missing strong correlation leads to computational artifacts. Let us briefly review this situation for the three fullerene isomers (C20, C36 = 36:15, and C60 = 60:1812) discussed broadly in the literature, as described succinctly in the previous paragraph. Our DFTB calculations correctly predict that both C20 and C60 have closed-shell ground state with the appropriate Ih point group symmetry, as expected. No DFTB wave function analysis has been performed for these wave functions, as the DFTB model does not use explicitly spin degrees of freedom (needed for studying the RKS/UKS instabilities) in its internal structure. For C36 = 36:15, DFTB predicts a closed-shell singlet with considerable HOMO-LUMO gap of 6.44eV. The missing strong correlation, known and broadly discussed in the literature, manifests itself in our calculations in quite surprising way. Namely, the 36:15 isomer is not the ground state of C36, with the 36:14 isomer lying 0.5 kcal/mol lower than the 36:15 isomer. [From the point of view of the energy differences appearing between various isomers of C36 (this energy span for the 15 isomers of C36 is over 110 kcal/mol in our DFTB calculations), the difference of 0.5 kcal/mol simply signifies that the isomers 36:14 and 36:15 are quasi-degenerate, particularly taking into account that the accuracy of DFTB is probably within 5-10 kcal/mol.] The real ground state designation of 36:15 comes from additional non-dynamic (strong) correlation contributions from other determinants (as analyzed in detail in DOI 10.1021/ja983853o), and is most likely to be missed by all single-reference based techniques that miss strong correlation.  Fortunately, such a situation is quite exceptional and for all the isomers of C20-C50  only two isomers (36:15 and 44:37) seem to experience strong correlation in this way in noticeable manner (for detail again see DOI 10.1021/ja983853o).
  2. The second reason is associated with the fact that our DFTB calculations predict the lion's share of isomers to be closed-shell singlets with considerable HOMO-LUMO gaps, typical for well-defined, stable chemical molecules in their ground state, which are usually well-described by a single determinant wave functions, and hence are not considered to be strongly correlated. Only in 49 (out of 29767) cases, our calculations predict negligible or small HOMO-LUMO gaps, suggesting multireference character of the ground state wave functions (often referred to as "strong correlation" in the solid-state physics parlance).  We are aware that these 49 cases are most likely insufficiently described by DFTB and we gave our premonition about using this portion of results for definitive reasoning by stating in the text of the original version of our article that: "the DFTB solutions with fractional occupations are not rigorous and do not correspond to a well-defined spin quantum number, and hence should be treated with caution". Fortunately, such a situation concerns only a very small portion of isomers (<0.5%) and most of the isomers in question are located very high in energy comparing to the ground state isomer.

Note that our DFTB calculations have revealed another important aspect of the "strong correlation" problem for the fullerene isomers. Namely, many isomers geometries taken from the original library initially showed quasidegeneracies in the molecular orbital spectra, possibly associated with their multireference character. Only after many hundreds steps of geometry optimization DFTB was capable of discovering other geometry minima, usually much lower in total energy, that showed considerable HOMO-LUMO gaps and clear closed-shell characters. We anticipate that most of these geometries were previously unknown, as geometry optimization of fullerene cages is usually an expensive endeavour, unless somebody possesses a modest computational tool for such a purpose, such as DFTB. It is possibly misleading to call the geometrical transition to these structures as Jahn-Teller effect, as both the structures (of multi-reference character, higher in energy  and  of single-reference character, lower in energy) have usually only the C1 symmetry designation, so effectively no symmetry reduction occurs. We believe that the optimized DFTB geometries of the isomers should be used as initial guesses for any further DFT optimizations, as usually the DFT and DFTB structures are quite close to each other. 

We agree with the referee that using the phrase "random reasons" to explain the correlation coefficient R^2 of about half was not rather a poor choice on our side in the context of the discussion above. We have changed now the relevant sentence to "The R2 coefficients of 0.486 shows that approximately half of the statistical variance is explained by the predicted linear trend plotted in Fig. 6 and the other half cannot be inferred from the bond orders alone." The actual reasons for "the other half" are unknown to us and possibly include also the strong correlation effects mentioned by the referee.

Since we have selected "an open review", this part of the discussion will be available to the interested reader, so we are not going to repeat it extensively in the main body of the article. However, we agree with the referee that this aspect (i.e., strong correlation) was completely overlooked in the first draft of our paper, and to make up for this shortcoming, we have made some modifications to the text to accommodate this discussion in an effective way in the main text. Here is the list of particular changes that have been introduced in the text in this context:

  1. A number of new references have been added to the paper to illustrate the strong correlation problem. This includes all the references suggested by the referee (except for the first two, which described strong correlation in single isomers, but in crystals of C60 and solid films of C70) and a few other references cited in or citing these studies. This addition definitely upgrades the quality of our paper in our understanding.
  2. A brief discussion of the strong correlation problem has been included in our manuscript in the following form: "It is important to highlight that DFTB (similarly to DFT) might poorly describe isomers with a quasidegenerate ground state (i.e., states with a pronounced multireference character for which a single Slater determinant is a bad approximation). Various interesting and unexpected methodological problems might manifest themselves in this context [93 – 96 ].
    Fortunately, such strongly correlated states do not occur too often; the study of all isomers of the classical fullerenes C20–C50 by Fowler, Mitchell, and Zerbetto [97] shows that only two (out of 812) isomers—36:15 and 44:37—experience pronounced energy stabilization (>15 kcal/mol) in the approximate CISD calculations with four frontier orbitals, which suggests pronounced multireference character of the underlying wave functions. DFTB (and DFT) would overlook such strongly correlated states predicting their energies too high. For example, among the isomers of C36, DFTB predicts the isomers 36:14 and 36:15 within 0.5 kcal/mol (for an almost analogous DFT result, see [98]), but DFTB misses the fact that 36:15 is the ground state. The problem is rather serious, as there is no well-developed computational protocol for establishing univocally reliable energy ranking of fullerene isomers at their optimized geometries [ 99 ]. An obvious candidate for computing such an energy ranking of fullerene isomers would be the CASSCF/PT2 scheme (see for example [ 100 ]), but creating such a ranking would constitute a considerable computational effort and has not been yet performed in a systematic manner."
  3.  A brief discussion of previous DFTB results is given as the following fragment at the beginning of Section 2: "[...]; note that this method has been successfully used for this purpose before [ 75 , 76]. Note also that the isomer stability ranking has been performed for a large group of fullerenes (see for example Refs. [77], [78 ], and [79 ]), but the published results usually reported only the most stable isomers. Therefore, for the requirements of the current work, we have decided to recompute all the rankings again from the scratch using consistently the same quantum chemical method, DFTB."
  4. The following fragment has been added to the conclusion section: " This should not be surprising in the light of the earlier results reported by Paulus [142 ] for C20–C36, but final confirmation of these results may require additional CASSCF calculations to avoid artificial Jahn-Teller-like effects [ 93 ,94 ]."

We believe that all the changes and modifications appropriately address the comments of the referee. We take this occasion again to thank the referee for careful reading of our manuscript and interesting comments.

Round 2

Reviewer 1 Report

Comments and Suggestions for Authors

Well revised accordingly to the comments. Publication is recommended.